# Gold Nanoparticles-Functionalized Cotton as Promising Flexible and Green Substrate for Impedometric VOC Detection

**DOI:** 10.3390/ma16175826

**Published:** 2023-08-25

**Authors:** Silvia Casalinuovo, Daniela Caschera, Simone Quaranta, Virgilio Genova, Alessio Buzzin, Fulvio Federici, Giampiero de Cesare, Donatella Puglisi, Domenico Caputo

**Affiliations:** 1Department of Information Engineering, Electronics and Telecommunications, Sapienza University of Rome, Via Eudossiana 18, 00184 Rome, Italy; silvia.casalinuovo@uniroma1.it (S.C.); alessio.buzzin@uniroma1.it (A.B.); giampiero.decesare@uniroma1.it (G.d.C.); domenico.caputo@uniroma1.it (D.C.); 2Institute for the Study of Nanostructured Materials CNR-ISMN, Strada Provinciale 35d/9 00010, Montelibretti, 00010 Rome, Italy; simone.quaranta@cnr.it (S.Q.); fulvio.federici@cnr.it (F.F.); 3Department of Chemical Engineering, Materials and Environment, Sapienza University of Rome, Via Eudossiana 18, 00184 Rome, Italy; virgilio.genova@uniroma1.it; 4Division of Sensor and Actuator Systems, Department of Physics, Chemistry and Biology (IFM), Linköping University, Campus Valla, 58183 Linköping, Sweden; donatella.puglisi@liu.se

**Keywords:** gold nanoparticle (AuNP), volatile organic compound (VOC), cotton, impedance measurements, citrate, PVP

## Abstract

This work focuses on the possible application of gold nanoparticles on flexible cotton fabric as acetone- and ethanol-sensitive substrates by means of impedance measurements. Specifically, citrate- and polyvinylpyrrolidone (PVP)-functionalized gold nanoparticles (Au NPs) were synthesized using green and well-established procedures and deposited on cotton fabric. A complete structural and morphological characterization was conducted using UV–VIS and Fourier transform infrared (FT–IR) spectroscopy, atomic force microscopy (AFM), and scanning electron microscopy (SEM). A detailed dielectric characterization of the blank substrate revealed interfacial polarization effects related to both Au NPs and their specific surface functionalization. For instance, by entirely coating the cotton fabric (i.e., by creating a more insulating matrix), PVP was found to increase the sample resistance, i.e., to decrease the electrical interconnection of Au NPs with respect to citrate functionalized sample. However, it was observed that citrate functionalization provided a uniform distribution of Au NPs, which reduced their spacing and, therefore, facilitated electron transport. Regarding the detection of volatile organic compounds (VOCs), electrochemical impedance spectroscopy (EIS) measurements showed that hydrogen bonding and the resulting proton migration impedance are instrumental in distinguishing ethanol and acetone. Such findings can pave the way for the development of VOC sensors integrated into personal protective equipment and wearable telemedicine devices. This approach may be crucial for early disease diagnosis based on nanomaterials to attain low-cost/low-end and easy-to-use detectors of breath volatiles as disease markers.

## 1. Introduction

In the last decade, great efforts have been devoted to the detection and understanding of disease-related volatile organic compounds (VOCs) [1]. It has been well-documented that VOCs generated from cell metabolic activities can provide useful information on the health status of an organism and, therefore, be used as novel biomarkers for diagnostic purposes [2,3]. The detection of different types of diseases, including gastric or breast cancer, intestinal diseases, neurological diseases (Parkinson’s and Alzheimer diseases, multiple sclerosis), diabetes mellitus, and tuberculosis using VOCs has attracted increasing attention among scientists [4]. Due to their volatility and small molecular size, VOCs generated through the metabolism of cells are mostly secreted through the exhaled breath (34%), reflecting a quantitative–qualitative correlation with various kinds of diseases. For this reason, VOCs contained in breath have received an increasingly significant deal of attention [5].

Acetone (CH_3_COCH_3_) and ethanol (CH_3_CH_2_OH) are among the most easily detected probes for VOCs exhaled in breath and have been studied as potential biomarkers for metabolic diseases. Acetone is considered a probe for metabolic diseases [6]. In the case of high acetone blood levels, a sweet smell can be noticed in the breath [7]. In addition, abnormal acetone concentrations can be indicators of other medical conditions like liver diseases [8], heart failure [9], chemotherapy [10], infections [11], and lung cancer [12].

Ethanol is a product of food metabolism; its presence in the breath is primarily associated with fatty liver disease [13], diabetes mellitus [14], and lung cancer [15]. Moreover, being a metabolic product, it can be a good indicator of smoking habits [16] or of an “under the influence” condition [17].

Gas chromatography mass spectrometry (GC–MS), proton transfer reaction mass spectrometry (PTR–MS), selected ion flow tube–mass spectrometry (SIFT–MS), and laser photoacoustic spectrometry [18] are generally used as standard techniques for VOC detection and analysis. However, the high costs of equipment, time-consuming procedures, pre-concentration steps, and need of trained professionals associated with the application of these techniques are some bottlenecks that limit their usage in clinical settings [19]. In the past few years, many researchers have been focusing on finding alternatives to these standard techniques by targeting portability, cost-effectiveness, sensitivity, and ease of operation, among other important challenges [20].

Advances in VOC detection have been aided by the development of nanomaterial-based devices characterized by a high surface-to-volume ratio, fast response and recovery times, miniaturization and low power consumption, etc. [4,21,22].

In this context, we present a gold-functionalized cotton textile as a promising chemo-resistive system that is capable of changing its electrical resistance when exposed to certain analytes. Due to its flexibility, low cost, and low environmental impact, cotton fabric possesses numerous advantages over silicon and/or plastic films as a suitable substrate for sensor integration [23]. Its porous texture results in a higher surface-to-volume ratio than standard substrates, leading to a greater number of adsorption sites (outer/inner pores) that can interact with a broad range of analytes [24]. Furthermore, Au NPs are capable of promoting fast response and recovery times and parts-per-billion (ppb) detection limits, and they can be employed on both stiff and flexible materials [25,26].

Au NPs properties are strictly dependent on their size, morphology, density, and on the specific surrounding chemical environment. This work explores two different, well-established, and green approaches for Au NPs synthesis. In detail, polyvinylpyrrolidone (PVP) and trisodium citrate dihydrate (Figure 1) were used as reducing/stabilizing agents in Au NPs synthesis.

In this study, we performed optical, structural, and morphological characterizations to understand the relationship between synthetic parameters, structure/morphology, and the electrical response of Au NPs embedded in cotton. Furthermore, we carried out electrical characterization to investigate the system’s ability to detect simple VOC probe molecules (i.e., acetone and ethanol).

The proposed Au NPs-functionalized cotton is amenable to a promising chemo-resistive device, which is an alternative to traditional diagnostic tools, for VOC detection. This is integrable into the portable sensor, opening a new frontier of measurements [27,28]. This may allow for further checkups in clinical and remote settings, providing additional information on the patient’s health status.

## 2. Materials and Methods

### 2.1. Chemicals

Tetra–Chloroauric (III) acid trihydrate (HAuCl_4_·3H_2_O, 99.995%), Polyvinylpyrrolidone (PVP_10_, average mol wt 10,000), Sodium citrate tribasic dihydrate (ACS reagent, ≥99.0%), ethanol (98%), and acetone (ACS reagent, ≥99.5%) were purchased from Sigma –Aldrich and used without further purification. All aqueous solutions were prepared using deionized-distilled water. Cotton fibers (Cod.558−79) were purchased by RS components, U.K.

### 2.2. Synthesis of Gold Nanoparticles Solutions (AuNP)

Gold nanoparticles (Au NPs) were prepared according to two different synthetic routes. In the first case, PVP_10_ was employed as a stabilizer/reducing agent, according to the procedure reported in [29]. Briefly, 5 g of PVP_10_ were dispersed in 50 mL of H_2_O under stirring. Then, 2.5 mL of a HAuCl_4_ 3.8 × 10^−3^ M aqueous solution were added, and the resulting mixture was left under stirring for 24 h at room temperature. The appearance of a reddish coloration indicated the formation of gold nanoparticles (AuNP_PVP). The final pH was 6. Au NPs were also prepared using a citrate-based reduction synthesis, based on the Turkevich method [30]. Briefly, 1 mL of a HAuCl_4_, 3.8 × 10^−3^ M aqueous solution was added to 50 mL of H_2_O under stirring. The solution was heated to the boiling point under reflux. Thereafter, 5 mL of a 38.8 mM sodium citrate solution were added. The final solution was left under stirring for 10 min and cooled at room temperature. In 30 min, the solution color changed to black and then red, proving Au NPs formation (AuNP_Citr). For completing the reaction, the solution was left under stirring at room temperature for further 4 h. The final pH was 6.

### 2.3. Cotton Functionalization

Two cotton pieces (100 mm × 100 mm), previously cleaned in an acetone bath for 3 h and then dried at 50 °C for 15 min, were put in two crystallization pots. Each sample was then soaked into the Au NPs solutions (AuNP_PVP and AuNP_Citr) for 24 h. The functionalized cotton fabrics were left drying in air at room temperature for at least 5 h. The color of the cotton fabrics changed from white to pink/purple depending on gold content, morphology, and density of Au NPs on the tissue. Cotton fabrics were labeled as AuNP_PVP/COT and AuNP_Citr/COT.

### 2.4. Structural and Morphological Characterizations

UV–VIS absorbance spectra for Au NPs solutions were collected in the 300–800-nm wavelength range with a double beam spectrophotometer V-660 (Jasco, Tokyo, Japan), using a 1-cm-wide quartz cell. The reflectance of the functionalized cottons was measured by means of the same UV–VIS spectrophotometer equipped with a reflectance integration sphere (60 mm).

The Fourier transform infrared (FT–IR) spectra were acquired using an IR Prestige-21 spectrometer (Shimadzu Europa GmbH, Düsseldorf, Germany) equipped with an attenuated total reflectance (ATR) module. For Au NPs solutions, few drops of Au NPs were deposited on the ATR and left drying at room temperature before measurements. The measurements were carried out in the 400–4000-cm^−1^ range by dint of a ZnSe crystal with a 0.2 cm^−1^ resolution.

Raman analysis was conducted at room temperature with a Renishaw RM 2000 (Gloucestershire, UK) using an Ar^+^ laser (514.5 nm excitation line) equipped with a Peltier-cooled charge-coupled device (CCD) camera and a 50× objective Leica optical microscope.

Atomic force microscopy (AFM) investigations were performed by means of Dimension 3100 Atomic Force Microscope (Bruker, MA, USA) equipped with a NanoScope IIIa, controller (Veeco, Santa Barbara, CA, USA) operating in tapping mode. A ScanAsyst–Air (Bruker) probes were used for acquiring the images, in air, that were then elaborated using the software Gwyddion v2.51.

Scanning electron microscopy (SEM) analysis was performed with a MIRA_3_ (from TESCAN, Brno, Czech Republic) equipped with EDAX TEAM.

### 2.5. Impedance Measurements

Au NPs-functionalized cottons were electrically characterized by electrochemical impedance spectroscopy (EIS) using a combined potentiostat/galvanostat/ZRA Reference 3000 (Gamry Instruments, Warminster, PA, USA) equipped with an electrically shielded cage. Gamry Framework/EchemAnalyst software was used for the experimental measurements and post-processing analysis. Impedance investigations were conducted with AC V_rms_ = 10 mV, 0 V DC bias, sweep frequency from 10 mHz to 1 MHz (10 points/decade). Cotton fabric was cut into 2.5-cm × 3-cm pieces and he measuring equipment, using alligator cables, in a two-electrode configuration (sandwich capacitor with aluminum foils). In addition, to ensure the electrical contact, two aluminum foil electrodes were sandwiched between two glass slides (5 cm × 5 cm) and kept in place by a weight of 900 g. Ethanol and acetone aqueous solution (40% in volume) were prepared as VOC probes and sprayed on the textile at a fixed distance of 10 cm just before arranging the experimental set-up. The considered distance was optimized for the solvent droplets to cover the whole substrate. Equivalent circuit models were implemented to extrapolate salient electrical parameters from resistance and reactance data.

## 3. Results and Discussion

### 3.1. Structural and Morphological Characterizations

Figure 2 reports the UV–vis. absorbance measurements of the Au NPs solutions and Au NPs-coated fabrics. For AuNP_Citr, the maximum absorption peak is centered at about 529 nm, while the corresponding band for the AuNP_Citr/COT is slightly red-shifted at about 538 nm. A similar situation is observed for the PVP-based samples: Au band is visible at 537 nm in solution, whereas the same signal is shifted at 557 nm on the cotton fabric. As previously reported in the literature [31], by UV–vis data, the average size of Au NPs in solution was estimated to be approximately 15–20 nm, while, considering the observed red shift on the cotton, the agglomeration effect increased their size to about 30–40 nm. The UV–Vis analysis demonstrated that Au NPs were successfully transferred, without any relevant formation of macro aggregates, from the solutions to the substrate by dint of a simple soaking process. The amount of Au NPs deposited on cotton fabrics was estimated by UV–Vis absorbance comparison, and the relative Au NPs density was found to be about 5.1 µg/cm^2^ for AuNP_Citr/COT and 4.6 µg/cm^2^ for AuNP_PVP/COT.

The effectiveness of the dipping process was also verified by FT–IR (Figure 3a) and the Raman spectroscopy (Figure 3b).

Differences between FT–IR spectra of blank cotton and Au NPs functionalized-cotton (Figure 3a) were found to be negligible due to the overlapping of most major cotton bands. The peaks at 3400, 2900, and 1057 cm^−1^ were attributed to the cotton fabric [32]. When PVP was used as a reducing/stabilizing agent for Au NPs formation, characteristic absorption peaks of the pyrrolidinyl group of PVP, corresponding to the C–O and C–N vibrations in PVP, were found at 1654 cm^−1^ and 1282 cm^−1^, respectively [33]. However, when sodium citrate was used, the most relevant feature in AuNP_Citr/COT turned out to be the great enhancement in intensity for the band at 1026 cm^−1^, which was attributed to C–O stretching vibrations. Nevertheless, the presence of a low-intensity peak at about 1740 cm^−1^ assigned to the non-conjugated C=O stretching vibrations of the carbonyl functional groups bounded on Au NPs during citrate reduction confirms the success of the functionalization process [34]. Conversely, noticeable vibrational bands that are characteristic for cotton fibers can be observed at about 1374, 1099, 527, 450, and 378 cm^−1^ in the Raman scattering spectra (Figure 3b) of blank cotton. Some of these bands were assigned to vibrations of β-1,4-glycosidic ring linkages between D-glucose units in cellulose [32,35]. Au NPs are well-known for being widely used as active substrates for enhancing Raman signals due to their localized surface plasmon resonance (LSPR) effect [36]. The surface-enhanced Raman spectroscopy (SERS) effect is strongly dependent on the Au NPs morphology and their surrounding environment [37,38]. No remarkable differences could be observed in the Raman spectrum of AuNP_PVP/COT with respect to blank cotton. On the contrary, a great enhancement of several Raman bands is detectable for AuNP_Citr/COT; for example, those at 2130, 1935, 1563, 269, and 115 cm^−1^. The closer the Au NPs are, the stronger the SERS effect. Therefore, this phenomenon is amplified in the citrate case (i.e., no steric stabilization involved) where particles are quite close. To better understand the influence of the two different stabilizers/reducing agents in the Au NPs formation, AFM and SEM analyses were performed. AFM images are shown in Figure 4. The blank cotton surface (Figure 4a) shows the typical fibrous morphology in which the single fibers are clearly visible. In AuNP_Citr/COT (Figure 4b), the surface appears to be rougher, and Au NPs of about 30–40 nm in diameter are present. Nevertheless, the peculiar fibrous structure of the cotton is retained. However, PVP polymer formed a plastic-like structure when dried. Therefore, for AuNP_PVP/COT, the inner structure of cotton was uniformly covered by a coating of PVP matrix in which Au NPs of about 30–40 nm, visible as small aggregates, were embedded (Figure 4c). These findings are consistent with the UV–Vis and FT–IR/Raman measurements.

The different morphology of the samples was also verified by SEM measurements (Figure 5) that showed a very homogeneous distribution of 40 nm of Au NPs on cotton fiber for AuNP_Citr/COT (Figure 5b). However, 60-nm-wide aggregates of small Au NPs (about 30–40 nm) incorporated in a sort of film framework are clearly visible for AuNP_PVP/COT (Figure 5c). Thus, Au NPs clusters tend to be isolated from within the insulating PVP matrix. Conversely, citrate-functionalized samples benefited from the citrate ability to disperse Au NPs by charge repulsion, which is a very effective approach in polar solvents (like water).

In order to comprehend the chemical composition of the functionalized cottons and assess the absence of interference in the electrical properties, energy-dispersive spectroscopy (EDS) measurements (Appendix A Appendix A) were also performed. The investigation revealed that the PVP-based sample has a chemical composition essentially consisting of oxygen, carbon, and gold. In the case of citrate-based cotton, EDS analysis also revealed the presence of sodium.

### 3.2. Electrical Measurements

The main purpose of a stabilizing agent is to control the growth of NPs and avoid agglomeration. Furthermore, the stabilization mechanism, the surrounding chemical, morphological characteristics, and solvent-related effects can differently affect the interaction between Au NPs and the specific analyte, inducing a change in the electrical response of the system. PVP is a biocompatible, amphiphilic, and water-soluble polymer extensively used in pharmaceutical, cosmetic, and food products. The hydrophobic polyvinyl chain acts as a tail group, whereas the highly hydrophilic pyrrolidone chain containing a double covalent carbon–oxygen bond (C = O) and nitrogen (N) act as a head group [29,39]. The interaction between PVP and Au surface derives from the donation of electrons from N or O in the side chain of the PVP to the Au NPs, leading to a concurrent steric and electrostatic stabilization. Nonetheless, steric stabilization is often regarded as the sole contribution to nanoparticle stabilization. Trisodium citrate dihydrate provides negatively charged citrate ions in solution, which absorb onto the Au NPs, acting as a “capping” agent. The final effect is to create an overall negative charge (i.e., electrical double layer) around the Au NPs that prevents further agglomeration [34].

We investigated the electrical behavior of Au NPs-coated cotton fabrics by means of impedance spectroscopy (EIS) to assess the effect of these different stabilizing agents for Au NPs.

A comparison of the impedance (Nyquist plots) of blank and Au NPs functionalized cottons is reported in Appendix A. Clear differences in terms of both impedance components (real and imaginary) values and curve shape were revealed. In particular, from Appendix A, a decrease in the overall impedance is observed as a consequence of AuNPs addition, with respect to blank cotton in both PVP and citrate fabrics. The greater resistance decrease measured in the AuNP_Citr/COT can be ascribed to better particle interconnection and the absence of an insulating polymeric matrix. AuNP_Citr/COT and AuNP_PVP/COT samples showed similar (sizeable) impedance components, suggesting that the porous insulating cotton texture mainly contributes to the overall impedance. Figure 6 shows the Bode plots of non-sprayed Au NPs functionalized cottons. Figure 7 displays the dielectric parameters calculated from the impedance data as reported in [40,41,42,43,44,45,46,47,48]. Both real (ε′) and imaginary (ε″) parts of permittivity increase in the low-frequency range (f < 10 kHz) because of “internal” mesoscopic interfacial polarization at the boundary between functionalized Au NPs and the cotton fabric.

Dielectric relaxation of the mesoscopic interfacial polarization can be appreciated at low frequencies. In fact, a peak in the dissipative component of complex permittivity (imaginary part ε″ and loss tangent tan(δ)) is observed around 1 Hz and 100 mHz for AuNP_Citr/COT and AuNP_PVP/COT, respectively. Moreover, an inflection point (at the frequencies corresponding to the peaks in the imaginary part of permittivity) was detected in the real (i.e., storage) part of the complex permittivity. PVP adsorption (functionalization) at the AuNP/fabric interface may justify lower relaxation frequencies (longer relaxation times) related to the “bulky” polymeric nature of the dispersant/reductant. Further relaxation peaks with a broad distribution reflecting the heterogeneity of the system were observed for both samples in the 1–100 Hz region.

The dependence of AC conductivity (σ_ac_) on frequency (f) (Figure 8) could be fitted (as most of disordered nanocomposite materials) with a Jonscher’s-like power law describing a universal dielectric response [49]:σ_ac_ = σ_dc_ + Aω^s^(1)
where ω (2πf) is the angular frequency; σ_dc_ is the conductivity for ω→0; A is the pre-exponential factor depending on frequency and temperature and related to the strength of the polarizability; and s is an exponent (0 ≤ s ≤ 1) that represents the level of interaction between charge carriers and the “lattice” around them.

AC conductivity dependence on frequency can be explained in terms of polarization effects resembling the behavior of lossy nanocapacitors composed of conductive clusters (Au NPs functionalized with citrate and/or PVP) and a dielectric matrix (the cotton fabric). As the frequency increases, nanocapacitors’ contribution to σ_ac_ becomes more important (i.e., the current flowing through the nanocapacitors rises) and the AC conductivity increases [50,51]. On a mesoscopic scale, a high-frequency electric field may promote fast oscillations of Au NPs surface electrons facilitating their tunneling between adjacent Au NPs [52]. Not surprisingly, fitting the data according to Jonscher’s model, the PVP-functionalized sample presented a lower σ_dc_ (4.91 × 10^−5^ S/cm) than the citrate-functionalized one (1.30 × 10^−4^ S/cm), in agreement with the larger real impedance component for f → 0 (see Figure 6). Most likely, smaller conductivity in the low frequency limit is probably due to PVP polymer chains acting as steric stabilizers for Au NPs. Consequently, PVP polymer chains kept Au NPs from interconnecting (uniformly), as verified by the AFM and SEM images (Figure 4c and Figure 5f). Moreover, the lower Au NPs surface density of the AuNP_PVP/COT needs to be considered. Low σ_dc_ values are consistent with low aspect-ratio nanostructures (Au NPs) at concentrations below the percolation threshold [53]. *s* coefficients were found to be between 0 and 1 (0.94 and 0.99 for the citrate and PVP sample, respectively) as predicted by Jonscher’s model. Although *s* versus temperature measurements would be required to establish the exact conduction mechanism [47], it can be stated that the higher *s* values for the PVP-functionalized sample are associated with polarization mechanisms (i.e., mobility of dipoles) extending over wider spatial regions like dipoles along lengthy polymer chains [54].

An electrical modulus analysis (Figure 9) was performed to remove electrode polarization effects (as opposed to mesoscopic interfacial polarization) [55,56] caused by the sample-to-electrode interface (i.e., sample/Al electrodes interface).

M′ (real component), M″ (imaginary component), and M* (complex electrical modulus) versus frequency graphs are reported in Figure 9. M′ and M″ approached almost zero as the frequency decreased. Therefore, electrode polarization contribution can be considered negligible [57,58,59]. Furthermore, M′ saturates to the M* value (3.5 and 5.5 for AuNP_PVP/COT and AuNP_Citr/COT, respectively) as the frequency increases, manifesting the highly capacitive nature (i.e., mesoscopic interfacial polarization and dipolar polarization) of the samples. The transition between long-range (low-frequency) and localized carrier conduction (high-frequency) can be deduced from the M″ versus frequency peak.

Electrical responses of the Au NPs-functionalized cottons were also investigated by using 40% of water solutions of ethanol and acetone as model probes for VOCs. The corresponding measurements for blank cotton were also conducted for comparison, and the results were shown in Appendix A. Marked differences in terms of impedance (for both real and imaginary components) before/after functionalizing the cotton fabric with the gold nanoparticles were found. The Au NPs presence determined lower impedance values, representative of a higher conductivity, with a shift of almost one order of magnitude compared to blank cotton. In the citrate sample, the major impedance decrease was due to better particle interconnection (i.e., shorter distance between gold nanoparticles clusters). Furthermore, the interfering effect of the solvent (i.e., water) or physiological solution on the electrical behavior of Au NPs-functionalized cottons was evaluated by specific experiments. The results (Appendix A) highlight a remarkably different behavior in terms of values range and curve shape. Upon distilled water spraying, blank cotton overall real component of the impedance drops of two orders of magnitude. Conversely, spraying distilled water on functionalized fabrics resulted into a more significant lowering of the overall resistance of the system. Indeed, the total real impedance of water-sprayed AuNP_PVP/COT was around 10^7^ Ω, whereas water-sprayed AuNP_Citr/COT displayed a value around 10^6^ Ω.

In addition, tests with 1% NaCl solution were performed (Appendix A). Clearly, water always exerts an effect on the electrical properties of the material. Nonetheless, a different behaviour, probably due to the different conduction mechanism, was found for the NaCl solution with respect to ethanol and/or acetone sprayed samples. In fact, the total resistance of NaCl-sprayed sample is at least one order of magnitude less than pure water and four orders of magnitude smaller than blank AuNPs-functionalized samples.

Equivalent circuit models (Appendix A) were devised and implemented to extrapolate physicochemical parameters of functionalized cotton fabrics (Figure 10) with and without interaction with the analytes (acetone or ethanol) [59]. A Bisquert-like universal transmission line was used to account for the porous cotton fabric substrate (medium-frequency part). A two-time constant circuit, comprised of two parallel non-ideal (i.e., constant phase element or CPE instead of ideal capacitance) RC elements, was employed in conjunction with the transmission line. A first R–CPE circuit (high-frequency contribution) belongs to the “geometric” capacitance of the measuring configuration (i.e., sample sandwiched between Al electrodes), while a second R–CPE represents the interfacial polarization of functionalized Au NPs embedded into the fabric (low-frequency part).

Equivalent circuits for the “sprayed” samples were developed by adding specific components to the original PVP and citrate models (Appendix A). The ethanol spraying implies a decrease of the total resistance of the systems (Figure 10a,c,d,f). A Grotthus-like proton “jumping” conduction mechanism [60], mediated by hydrogen bonding with the PVP- and Citrate-capped Au NPs [61,62], may be responsible for the higher charge transport rate (i.e., lower overall resistance) caused by the ethanol/water solution. Nonetheless, a low-water ionic product (K_w_ = 10^−14^) and ethanol small acid constant (pK_a_ = 16) still pinned the overall resistance in the tens of MΩ range.

However, (Figure 10a,b,d,e) a slight (citrate case) or negligible (PVP case) change in the real part of the impedance was revealed for the acetone-sprayed AuNP_Citr/COT and AuNP_PVP/COT, probably because of partial water contribution. The total resistance (at low frequency) of the acetone-sprayed samples did not drop below 9 × 10^8^ Ω (almost two orders of magnitude higher than their ethanol-sprayed counterparts) due to the lack of protons available for H- bonding. As a consequence of H^+^ mobility, a new series, R–CPE, was added to consider ethanol/acetone adsorption on the Au NPs. Indeed, a non-ideal capacitance represents the interfacial capacitance (i.e., double layer-like) stemming from VOCs adsorption on Au NPs mediated by the citrate and/or the PVP, whereas resistance (R_23_ in Appendix A) describes proton migration through the Grotthus mechanism. Aluminum contacts can safely be considered “blocking electrodes”, and a charge transfer is not expected to take place at the fabric/Al interface. For the sake of clarity, experiments were conducted using copper foils instead of aluminum, maintaining the same configuration, and electrodes gained similar results (Appendix A). Proton migration resistance values reflected the different interaction between the analyte and substrate. For example, when ethanol was sprayed on the functionalized cotton, R_23_ was 5.54 × 10^3^ Ω and 1.69 × 10^6^ Ω for AuNP_Citr/COT and AuNP_PVP/COT, respectively. The corresponding resistance values for acetone-sprayed samples were 9.64 × 10^5^ Ω (Citrate) and 4.27 × 10^8^ Ω (PVP). The observed electrical properties could be related to the specific chemical environmental and stabilizing agent for Au NPs. Due to the special molecular structure of PVP, the carbonyl functional group can stably complex the Au Nps [63]. However, the presence of polar lactam groups can capture polar molecules by hydrogen bonding [64]. In the case of citrate, instead, electrostatic attraction between the citrate anions on Au NPs and the positively charged groups of probes may drive the molecules to approach AuNP surfaces, thus facilitating their adsorption [65]. In this way, ethanol’s capability to form H-bonds with both the functionalized substrate seems to be a reasonable explanation for smaller impedance values. Certainly, ethanol can form H-bonds with both citrate and PVP (through ionized carboxyl groups and carbonyl/N-carrying rings, respectively) allowing proton transfer and subsequent H^+^ transport to occur. As stated previously, a larger charge transport resistance of the PVP samples is justified by the intrinsic polymeric nature of the binder, creating a larger spacing between Au NPs, as verified by AFM and SEM images (Figure 4c and Figure 5f). However, acetone forms no hydrogen bonds with the functionalized Au NPs and proton transport can only take advantage of a limited number of water molecules interacting with the substrate. Concerning the capacitive component related to the VOCs adsorption on Au NPs, CPE magnitude was found to be higher when ethanol was sprayed on both citrate- and PVP-functionalized samples. Ethanol’s larger DC dielectric constant (i.e., ≈24 at 25°) compared to the acetone one (i.e., ≈21 at 25°), in conjunction with the stronger VOC-substrate interaction caused by hydrogen bonding (with respect to dipolar Van der Waals interactions responsible for acetone adsorption), can explain such findings.

## 4. Conclusions

This work presents promising results on the development of health-wise VOC sensors relying on Au NPs-functionalized cotton fabric. Specifically, gold nanoparticles (Au NPs) were synthesized by exploiting PVP or sodium citrate as reductants and stabilizers. Au NPs were transferred on a cotton substrate by dip-coating. The optical characterization in the UV–Vis range of both solution and coated cottons demonstrated the presence of Au nanoparticles with an approximately 20–40 nm in diameter.

The electrical characterization performed by EIS showed that the impedance behavior of the Au NPs functionalized cottons can be related to the nature of the specific reductant/stabilizer (citrate or PVP) used in the Au NPs synthesis, although some interfering effects originating from the high resistance of the cotton fabric substrate are observed. In particular, AuNP_PVP/COT demonstrated smaller DC conductivity with respect to AuNP_Citr/COT, which is probably ascribed to poor Au NPs interconnection along with lower AuNP surface density. Furthermore, when ethanol or acetone were sprayed on the functionalized cottons, the analytes adsorption resulted in transport of the H^+^ (stemming from ethanol and partially from water). H-bond was identified as a possible mechanism accounting for proton mobility (i.e., Grotthus-like transport). Therefore, ethanol’s ability to form hydrogen bonds with citrate and PVP can be exploited as a sensing principle affecting macroscopic electrical properties. On the contrary, the poorer ability of acetone to form hydrogen bonds with the functionalized Au NPs makes the electrical measurements less sensitive, since the proton transport can only take advantage of a limited number of water molecules interacting with the substrate. In this case, specific functionalizations on the Au NPs should be added to improve the acetone interaction with the substrate.

These results represent a first step towards the development of a simple, flexible, “low-cost/low-end” VOC detector based on impedance changes on a cotton substrate coated with Au NPs. Such a device could be instrumental in replacing blood-urine tests or biopsy as a non-invasive diagnostic tool. Moreover, lower costs, a quicker analysis time, and portability are the main advantages over traditional biomedical investigation methods.

## Figures and Tables

**Figure 1 materials-16-05826-f001:**
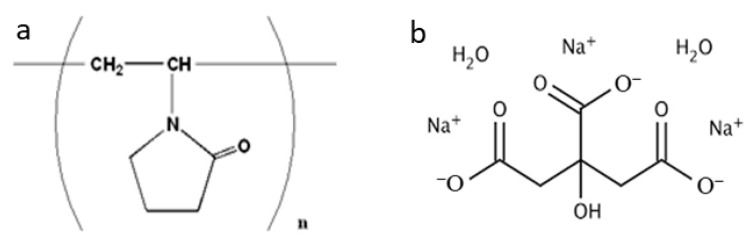
Chemical structure of (**a**) Polyvinylpyrrolidone (PVP) and (**b**) Trisodium citrate dihydrate.

**Figure 2 materials-16-05826-f002:**
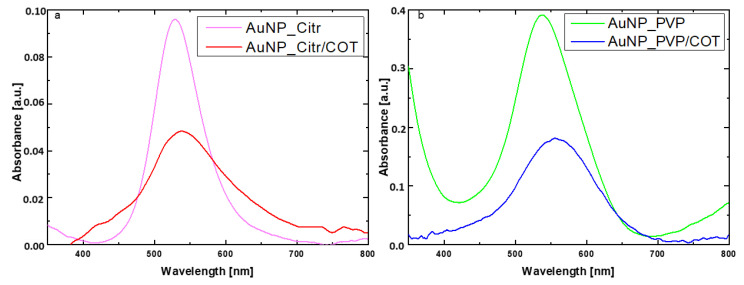
UV–Vis absorption spectra for (**a**) AuNP_Citr in solution and on cotton and (**b**) AuNP_PVP in solution and on cotton.

**Figure 3 materials-16-05826-f003:**
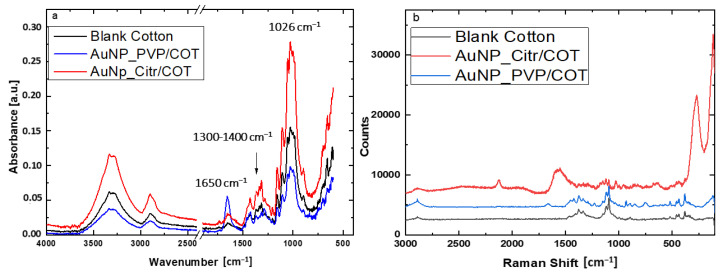
(**a**) ATR_FT–IR spectra and Raman spectra for blank cotton, (**b**) AuNP_PVP/COT and AuNP_Citr/COT.

**Figure 4 materials-16-05826-f004:**
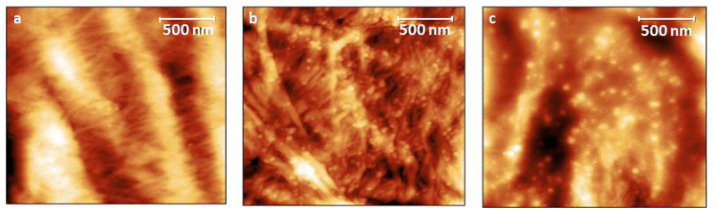
AFM images for (**a**) Blank cotton; (**b**) AuNP_Citr/COT; and (**c**) AuNP_PVP/COT.

**Figure 5 materials-16-05826-f005:**
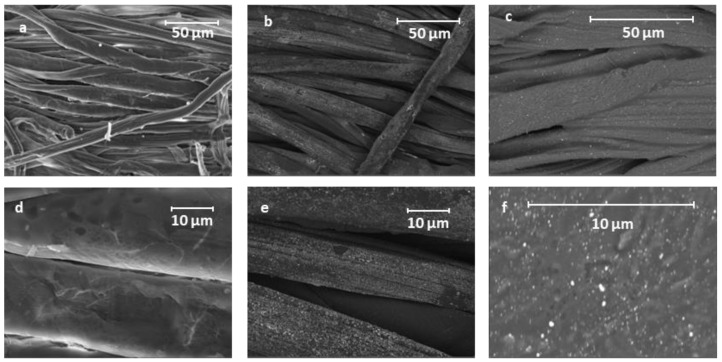
SEM images for blank cotton (**a**,**d**); AuNP_Citr/COT (**b**,**e**); and AuNP_PVP/COT (**c**,**f**). Images (**d**–**f**) refer to magnification of images (**a**–**c**), respectively.

**Figure 6 materials-16-05826-f006:**
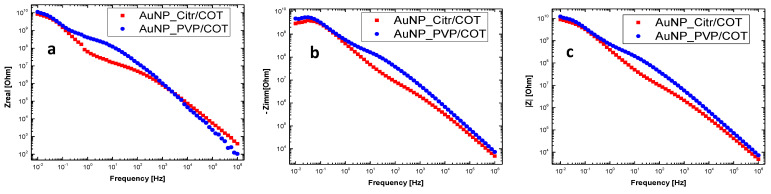
Impedance vs. frequency graph (Bode plots) of not-sprayed AuNP-coated cotton fabrics. (**a**) Real part (Z_real_); (**b**) Imaginary part (-Z_imm_); and (**c**) Magnitude |Z| of impedance are displayed.

**Figure 7 materials-16-05826-f007:**
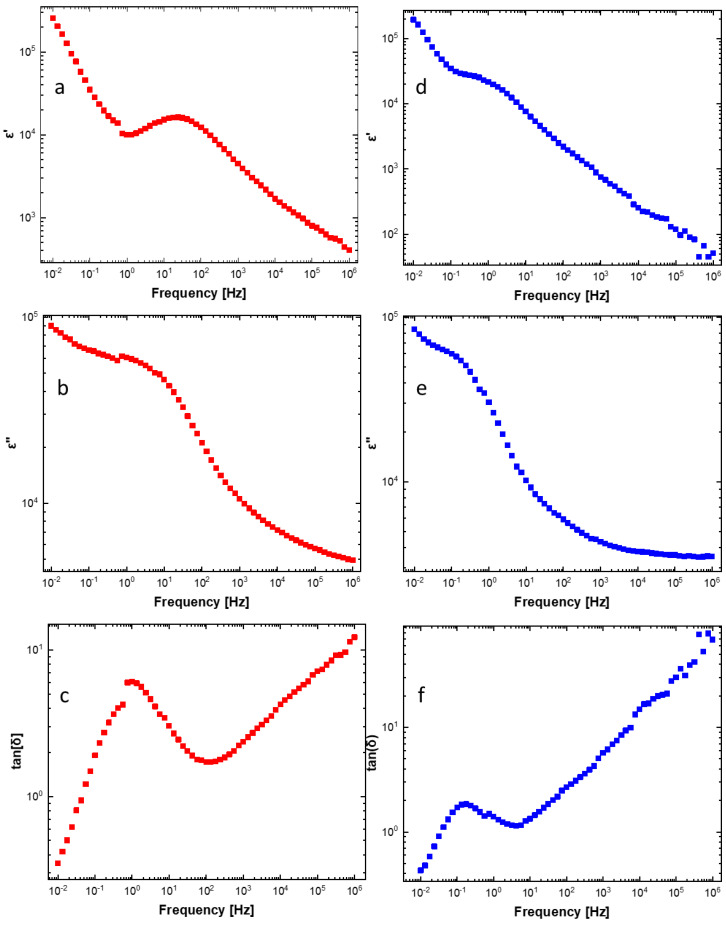
Dielectric permittivity parameters (tan(δ), ε′, ε″) of AuNP_ Citr/COT (**a**–**c**)—in red, and AuNP_PVP/COT (**d**–**f**)—in blue.

**Figure 8 materials-16-05826-f008:**
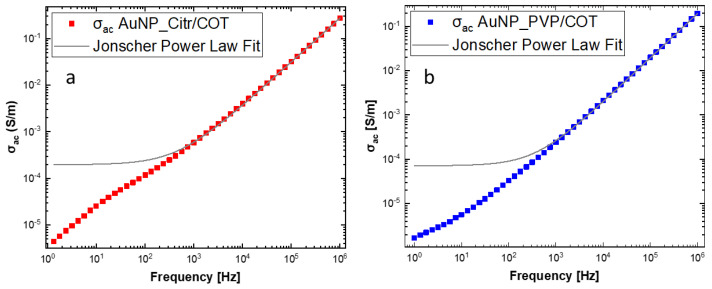
Dependence of AC conductivity (σ_ac_) on frequency for (**a**) AuNP_Citr/COT and (**b**) AuNP_PVP/COT.

**Figure 9 materials-16-05826-f009:**
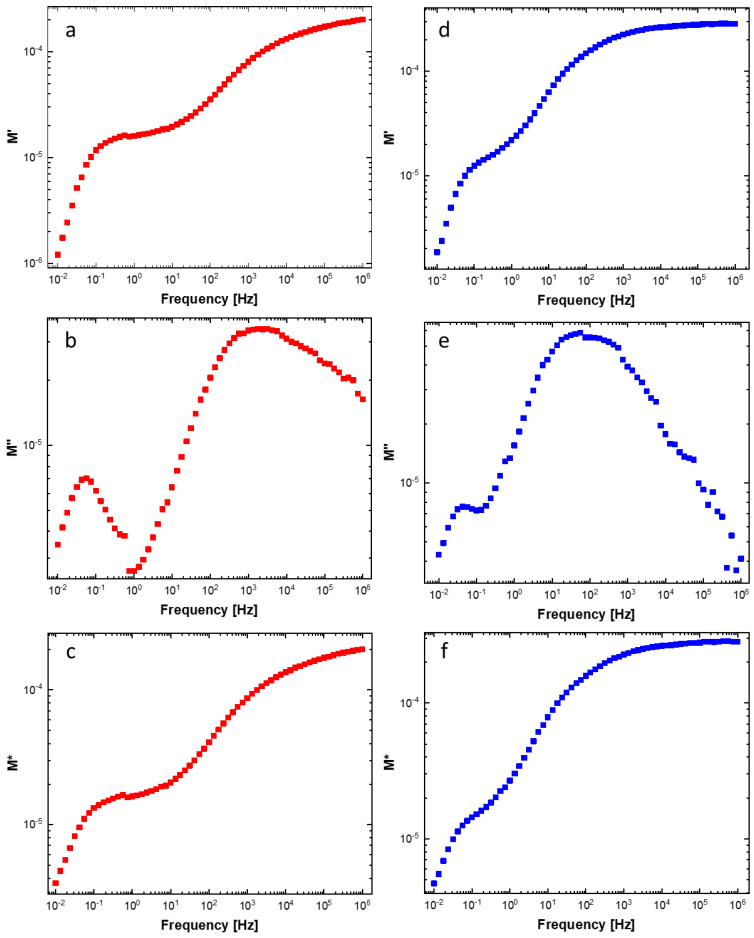
Dependence of electrical modulus parameters (M′, M″, M*) on frequency for (**a**–**c**) AuNP_Citr/COT (in red) and (**d**–**f**) AuNP_PVP/COT (in blue) cotton fabrics.

**Figure 10 materials-16-05826-f010:**
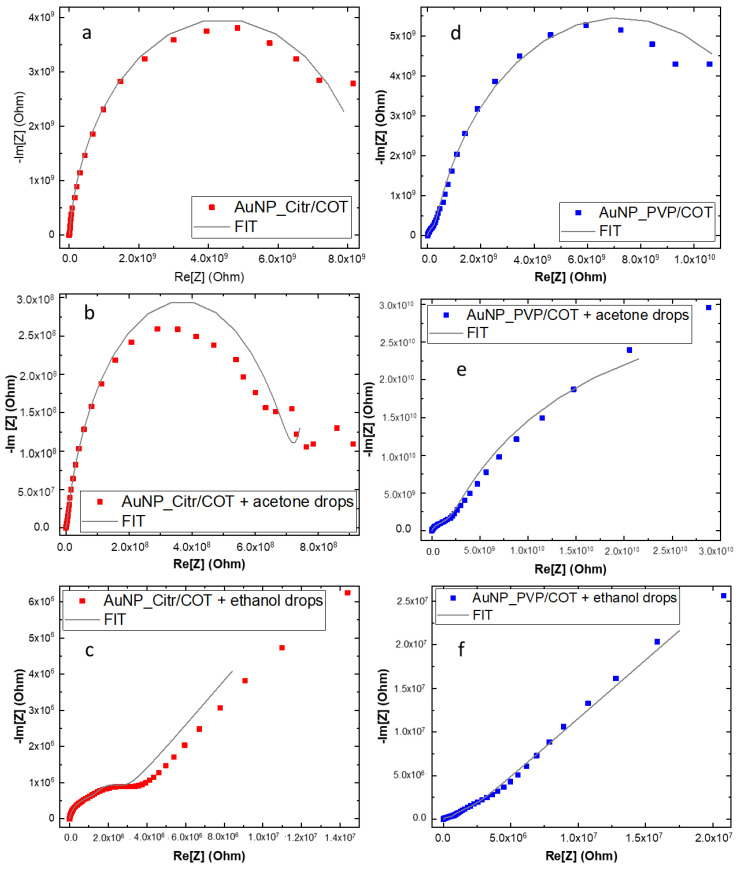
Nyquist plots of AuNP_PVP/COT (**a**–**c**) and AuNP_Citr/COT (**d**–**f**) before/after VOC solutions spraying.

## Data Availability

Not applicable.

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
