# Peer review of "Gold Nanoparticles-Functionalized Cotton as Promising Flexible and Green Substrate for Impedometric VOC Detection"

_materials, 2023, doi:10.3390/ma16175826_

Round 1
Reviewer 1 Report
The manuscript describes the assessment of Au nanoparticles coated cotton as an impedimetric sensor for VOCs presence. In my opinion the reasoning behind the presented measurements is not substantiated by the results by lack of the negative experiments. It is impossible to conclude from the presented results that the material works as designed because of three main gaps in the reasoning:
1. There is no comparison of the electric properties of blank cotton with Au nanoparticles coated cotton.
2. The response of the Au nanoparticles coated cotton is assessed using the water solution of alcohol and acetone. Such measurement doesn’t prove their sensitivity to VOCs unless they are compared to the response to pure water.
3. There is no comparison of the changes of electric properties of the Au nanoparticles coated cotton to the response obtained for blank cotton to water and water solution of VOCs.
Unless such experiments show significant differences between the negative and positive sample no conclusion may be drawn. Additionally:
4. One of the main issues of the sensors is the specificity of the response. Sensors is valuable if one can show lack of the sensitivity on other factors like, for example, temperature or other substances that may act on the sensors. Since the VOCs sensitive cotton would be excellent material for the intelligent clothing then it should not be sensitive to the sweat or at least NaCl solution. I think such experiment would also add to the value of the results and discussion.
Without addressing those issues I think this manuscript at current state is not suitable for publication.
Additional minor remarks:
5. Whenever there are plots of the electrical response that are more or less proportional or inversely proportional to logarithmically changing frequency, then the Y axis on the plot should also be logarithmic. Otherwise it is impossible to see any details at the lower parts of such graphs. From the quantities that were shown in the manuscript in most cases the Z, |Z|, conductance, sometimes also loss factor should be shown in log scale on Y axis.
6. Please replace figure 6 with regular Bode plot of the impedance spectra.
7. Figure 7a shows incorrect real permittivity evaluation. In that kind of the samples it is impossible to achieve permittivity smaller than 1 neither it should not increase with the increase of the frequency. This show both improper calculations and instability of the sample during the time of the measurement.
8. Figure 8 – the plots are completely unreadable due to the linear Y axis.
9. Figure 10 – the plots show that the model does NOT fit the data.
10. I cannot imagine how do you spray the cotton sample with water solution of VOCs if the cotton is sandwiched between two aluminium foil electrodes and two glass plates.
11. I don’t think the aluminium is the best choice for the electrodes. I suspect that a lot of the apparent ac properties come from the interface between the electrode and the cotton.
12. In my opinion the analysis of the ac response is weakly supported by the literature. Authors provide the citations substantiating their reasoning citing the papers which relate to another kind of samples. For example [44] shows ZnO-PVA composites, in which both phases are more or less dielectric, therefore Maxwell-Wagner polarisation can occur. If one phase of the composite is a conductor I would not expect such phenomenon. [45] does not discuss Jonscher power law, and so on. In my opinion most of the claims are unsubstantiated.
Author Response
We would like to thank the Reviewers for their time, suggestions and comments. We have revised the manuscript according to the Reviewers’ observations and we are confident the changes made will clarify their doubts.
REVIEWER 1
The manuscript describes the assessment of Au nanoparticles coated cotton as an impedimetric sensor for VOCs presence. In my opinion the reasoning behind the presented measurements is not substantiated by the results by lack of the negative experiments. It is impossible to conclude from the presented results that the material works as designed because of three main gaps in the reasoning:
- There is no comparison of the electric properties of blank cotton with Au nanoparticles coated cotton.
R: We are grateful to the reviewer for the comment. Impedance characterization of blank cotton has been added to SI (see Fig. S3). As it can be seen, there are clear differences in terms of both impedance components (real and imaginary) values and curve shape. In particular, from Fig. S3a and Fig. S3c, a lowering in the overall impedance is observed as a consequence of AuNPs addition with respect to blank cotton (black squares) in both PVP (red circles) and Citrate (blue triangles). The greater resistance decrease measured in the Citrate cotton at medium and high frequency can be ascribed to better particle interconnection and absence of insulating polymeric matrix. More explanations are added in the text.
- The response of the Au nanoparticles coated cotton is assessed using the water solution of alcohol and acetone. Such measurement doesn’t prove their sensitivity to VOCs unless they are compared to the response to pure water.
R: We thank the reviewer for bringing this point up. Impedance spectra of both functionalized fabrics (AuNP_Citr/COT and AuNP_PVP/COT) after only distilled water spraying have been added in the SI (Fig. S4). Acetone- sprayed samples show a bigger impedance compared to distilled water. In the PVP case (Fig. S4a and in the “blow-up” Fig S4b), real component of impedance reaches tens of GΩ in respect of water resistance values (MΩ). Similarly, impedance's imaginary component rises upon acetone spraying. Also in the citrate case (Fig. S4c and in the “blow-up” Fig S4d), increased values of the real and imaginary components can be seen with respect to those corresponding to water spraying. Such findings are consistent with replacement of a highly polar solvent (water) with a slightly polar solvent (acetone). In addition, higher resistance values are comparable with a lesser extent of H-bond formation (acetone solution contains 60% of water). Conversely, when ethanol was sprayed, a major impedance decrease occurred, compared to acetone. As for PVP , EtOH presence reduced system impedance (both -Im(Z) and Re(Z)) to the Ω-MΩ, showing an overall impedance on the same order of magnitude (107) of water. However overall resistance is almost double (20x106) than the water one (10x106). The text has been modified accordingly
Nevertheless, considering the obtained data, further studies are in progress for reducing water interference [as suggested in ref 30] and, therefore, improving material’s sensitivity.
- There is no comparison of the changes of electric properties of the Au nanoparticles coated cotton to the response obtained for blank cotton to water and water solution of VOCs.
R: Thanks to the reviewer for these correct observations. As requested, we have conducted the suggested measurements and reported the experimental results in the SI (Figure S5).
Upon distilled water spraying, blank cotton overall real component of the impedance (brown star curve) drops of two orders of magnitude, with respect to the non-sprayed sample. Conversely, spraying distilled water on Au NPs functionalized fabrics resulted into a more significant lowering of the overall resistance of the systems. Indeed, water-sprayed AuNP_PVP/COT showed an overall real impedance around 107, whereas water-sprayed AuNP_Citr/COT displayed a total resistance value around 106. On the other hand, when volatiles were sprayed on blank fabric impedance rose to hundreds of gigaohm, which was very hard to appreciate with the FRA equipment. On the contrary, both functionalized samples presented a completely different data range than the AuNP_PVP/COT (around 2,5x1010 and 2x107 for acetone and ethanol drops, respectively) and the AuNP_Citr/COT (around 8x108 and 1,4x107 for acetone and ethanol drops, respectively). The text has been modified accordingly.
Unless such experiments show significant differences between the negative and positive sample no conclusion may be drawn. Additionally:
- One of the main issues of the sensors is the specificity of the response. Sensors is valuable if one can show lack of the sensitivity on other factors like, for example, temperature or other substances that may act on the sensors. Since the VOCs sensitive cotton would be excellent material for the intelligent clothing then it should not be sensitive to the sweat or at least NaCl solution. I think such experiment would also add to the value of the results and discussion.
R: Thanks to the reviewer for this useful comment. We completely agree with him/her that the most important behaviour for a sensor is its selectivity and avoiding imterference from biological fluid should be one issue in sensorig measurements. Hovewer, the aim of this work is not the fabrication of an optimized cotton-based sensor. Indeed, the scope of the paper is to explore different synthetic approaches in gold nanoparticles fabrication, to understand the influence of the stabilizing agents on the electrical properties of the functionalized cotton. For clarity, we modified the paper title as “Gold nanoparticles functionalized cotton as promising flexible and green substrate for impedometric VOCs detection”.
According to the reviewer suggestion, additional, tests with 1% NaCl solution were performed (Fig. S6 of the SI). Even if an effect on the electrical behaviour of the materiali s observable, probably due to the different conduction mechanism for the NaCl solution the total resistance of NaCl-sprayed sample is at least 1 order of magnitude less than pure water and 4 orders of magnitude smaller than blank AuNPs-functionalized samples.
In addition,we have already planned more experimental tests, in order to check the potentialities of our system, controlling the temperature. Furthermore, specific surface treatments will be evaluated in order to overcome the issues related to the water interference (making more hydrophobic the cotton but leaving it permeable enough to polar molecules like ethanol or biological macromolecules).
Without addressing those issues I think this manuscript at current state is not suitable for publication.
Additional minor remarks:
- Whenever there are plots of the electrical response that are more or less proportional or inversely proportional to logarithmically changing frequency, then the Y axis on the plot should also be logarithmic. Otherwise it is impossible to see any details at the lower parts of such graphs. From the quantities that were shown in the manuscript in most cases the Z, |Z|, conductance, sometimes also loss factor should be shown in log scale on Y axis.
R: Thank you for this comment. All the required graphs have been replotted in a log scale.
- Please replace figure 6 with regular Bode plot of the impedance spectra.
R: Thank you for this comment. Figure 6 has been replaced as requested by the reviewer.
- Figure 7a shows incorrect real permittivity evaluation. In that kind of the samples it is impossible to achieve permittivity smaller than 1 neither it should not increase with the increase of the frequency. This show both improper calculations and instability of the sample during the time of the measurement.
R: Thank you for noting this incongruence. We apologize for the mistake, due to a conversion error regarding the surface area of the sample. Right conversion has been applied and the corresponding new data have presented in the plot.
- Figure 8 – the plots are completely unreadable due to the linear Y axis.
R: Thanks for this comment. The plot have been modified according to a logarithmic scale to improve data readability as requested.
- Figure 10 – the plots show that the model does NOT fit the data.
R: Thanks for this useful comment. We have carefully checked the data and they tried to apply the model that reflected best the real physical behaviour of the systems. In these conditions, we have considered as the best fits the ones with a χ2 value lower than 10-3, that is considered the minimum reference value for acceptable measurements.
- I cannot imagine how do you spray the cotton sample with water solution of VOCs if the cotton is sandwiched between two aluminium foil electrodes and two glass plates.
R: Thanks for this comment. The acqueous probe solution was sprayed just before preparing the measurement configuration. The operation takes few seconds, since the procedure has been well estabilished, prepared and tested several times.
- I don’t think the aluminium is the best choice for the electrodes. I suspect that a lot of the apparent ac properties come from the interface between the electrode and the cotton.
R: Thanks for this comment. Aluminium is supposed to be passivated and allows no charge transfer at the sample electrode interface. Furthermore, Al contacts can be considered inert since in the solutions there are no chemicals that can react with aluminium. For clarity, experiments have been replied using copper foils (Fig.S7) and no substantial differences are observed. Besides, electric module plots show that electrode/sample interfacial polarization effect can be negletcted as reported in literatrure (see ref 56)
- In my opinion the analysis of the ac response is weakly supported by the literature. Authors provide the citations substantiating their reasoning citing the papers which relate to another kind of samples. For example [44] shows ZnO-PVA composites, in which both phases are more or less dielectric, therefore Maxwell-Wagner polarisation can occur. If one phase of the composite is a conductor I would not expect such phenomenon. [45] does not discuss Jonscher power law, and so on. In my opinion most of the claims are unsubstantiated.
R: Thanks the reviewer for this observation. Even if the Maxwell-Wagner-Sillar polarization has been already reported for gold nanoparticles disperse in an insulating matrix in several papers (see ref. 39-42), we agree that our definition of “Maxwell Wagner polarization” maybe has been improperly used, creating some misunderstanding.
With the “Maxwell Wagner polarization”, we referred to the double layer capacitance at the gold nanoparticles/stabilizer/cotton interface. In other words, we meant to describe the formation of nanoscopic capacitors where gold nanoparticles act as the metallic plates and the absorbed polymer or citrate molecule in conjuction with the cotton fabric act as the dielectric of such capacitors [see ref 43,44]. Conductivity graphs have been replotted in a log log scale and Jonsher law fit has been performed again limiting the frequerncy range to 1 Hz (Figure 8). As expected the conductivity differences between citrate and PVP remain. Nonetheless, we would like to point out that linear y scale for conductivity measurements have been reported on literature for gold nanoparticles [see ref. 45-47]. Therefore, our conductivity data are consistent with data reported in literature concerning gold nanoparticles embedded in a polymer/insulating matrix. In fact, in such cases conductivity decreases below 1Hz. The text has been accordingly modified and the references changed.
Reviewer 2 Report
Report on the manuscript materials-2535361 entitled “Gold nanoparticles functionalized cotton as promising flexible and green impedance sensor for VOCs detection”.
The submitted manuscript should be revised. In this work, application of gold nanoparticles on flexible cotton fabric as acetone and ethanol sensing device by impedance measurements. Some structural and morphological characterization were carried out which revealed interfacial polarization effects related to both AuNPs and their specific surface functionalization. In addition, the following points should be addressed:
1. The language of the manuscript should be revised.
2. “AuNPs” should be “Au NPs” in all parts of the manuscript.
3. In keywords: “Impedance sensor” should be “Impedance measurements.”
4. In introduction, so many paragraphs are present and could be combined so, I suggest dividing the introduction part to only 3 major paragraphs to be easily understood.
5. More details of experimental work should be added. For example, “The appearance of a reddish coloration indicated the formation of gold nanoparticles (AuNP_PVP).” the pH and time of color appearance should be added.
6. The authors write “while the corresponding band for the AuNP_Citr/COT is slightly red-shifted at about 538 nm”, why?
7. In FT-IR, the peak at 1652 cm-1 was attributed to C=O and the C=O and the reported peaks of carbonyl groups are around 1700-1750 cm-1. Please explain?
8. The paragraph explained the equivalent circuit parameters (before figure 10) should have suitable references [suggested references: Solid-state electronics 99 (2014): 84-92. & Journal of Alloys and Compounds 816 (2020): 152513].
9. Conclusion should be clearer, and it should have clearly what achieved in this work.
Needs major revision in English quality.
Author Response
We would like to thank the Reviewers for their time, suggestions and comments. We have revised the manuscript according to the Reviewers’ observations and we are confident the changes made will clarify their doubts.
REVIEWER 2
Report on the manuscript materials-2535361 entitled “Gold nanoparticles functionalized cotton as promising flexible and green impedance sensor for VOCs detection”.
The submitted manuscript should be revised. In this work, application of gold nanoparticles on flexible cotton fabric as acetone and ethanol sensing device by impedance measurements. Some structural and morphological characterization were carried out which revealed interfacial polarization effects related to both AuNPs and their specific surface functionalization. In addition, the following points should be addressed:
- The language of the manuscript should be revised.
R: The manuscript has been fully revised and english form improved
- “AuNPs” should be “Au NPs” in all parts of the manuscript.
R: Thanks to the reviewer for these correct observations The change is done in all the parts of manuscript.
- In keywords: “Impedance sensor” should be “Impedance measurements.”
R: Thanks to the reviewer for these correct observations. Impedance sensor has been changed as impedance measurements
- In introduction, so many paragraphs are present and could be combined so, I suggest dividing the introduction part to only 3 major paragraphs to be easily understood.
R: Thanks for this useful comment. The introduction has been fully revised and organized, according to the reviewer suggestion.
- More details of experimental work should be added. For example, “The appearance of a reddish coloration indicated the formation of gold nanoparticles (AuNP_PVP).” the pH and time of color appearance should be added.
R: Thanks to the reviewer for this comment. More details about the experimentals have been added in the text.
- The authors write “while the corresponding band for the AuNP_Citr/COT is slightly red-shifted at about 538 nm”, why?
R: Thanks to the reviewer for this comment. Gold nanoparticles in solution present a typical plasmonic band at around 520nm, but it is strongly dependant to their size, to the specific solvent or to the presence of stabilizing agent. Also, when the gold solution is deposited onto a substrate, some agglomeration between smallest nanoparticles in bigger cluster could occur, leading to a red shift in the gold plasmonic band. In our case, for both the samples on cotton a red shift is observed, with respect to the Au solutions, attributed to the formation of larger clusters or particles. This effect is confirmed by SEM and AFM measurements, in which the size of Au NPs is reported to be around 40-60nm. Some changing in the text are adding for clarification.
- In FT-IR, the peak at 1652 cm-1 was attributed to C=O and the C=O and the reported peaks of carbonyl groups are around 1700-1750 cm-1. Please explain?
R: Thanks to the reviewer for this comment. For a mistake, the peak at 1652cm-1 was attributed to the carbonyl groups. Instead the band at 1652cm-1 for PVP are normally referred to the stretching vibration of the C-O in the pyrrolidinic group, as reported in ref.37. The mistake have been corrected.
- The paragraph explained the equivalent circuit parameters (before figure 10) should have suitable references [suggested references: Solid-state electronics 99 (2014): 84-92. & Journal of Alloys and Compounds 816 (2020): 152513].
R: We are grateful to the reviewer for the comment. The paragraph has been revised and the references suggested by the reviewer added.
- Conclusion should be clearer, and it should have clearly what achieved in this work.
R: Thanks to the reviewer for the comment. The conclusion paragraph has been revised, making clearer the aim of the work and the results obtained.
- Comments on the Quality of English Language
Needs major revision in English quality.
R: English has been fully revised and improved
Round 2
Reviewer 2 Report
Accepted